# *In silico* Analysis of Genetic Diversity of Human Hepatitis B Virus in Southeast Asia, Australia and New Zealand

**DOI:** 10.3390/v12040427

**Published:** 2020-04-09

**Authors:** Ngoc Minh Hien Phan, Helen Faddy, Robert Flower, Kirsten Spann, Eileen Roulis

**Affiliations:** 1School of Biomedical Sciences, Faculty of Health, Queensland University of Technology, Kelvin Grove, Queensland 4059, Australia; hfaddy@usc.edu.au (H.F.); kirsten.spann@qut.edu.au (K.S.); 2Research and Development Australian Red Cross Lifeblood, Kelvin Grove, Queensland 4059, Australia; rflower@redcrossblood.org.au (R.F.); eroulis@redcrossblood.org.au (E.R.); 3School of Health and Sport Sciences, University of Sunshine Coast, Petrie, Queensland 4502, Australia; 4Institute for Health and Biomedical Innovation, Queensland University of Technology, Herston, Queensland 4006, Australia

**Keywords:** human hepatitis B virus, genetic diversity, genotypes, serotypes, HbsAg subtypes, recombination, mutations

## Abstract

The extent of whole genome diversity amongst hepatitis B virus (HBV) genotypes is not well described. This study aimed to update the current distribution of HBV types and to investigate mutation rates and nucleotide diversity between genotypes in Southeast Asia, Australia and New Zealand. We retrieved 930 human HBV complete genomes from these regions from the NCBI nucleotide database for genotyping, detection of potential recombination, serotype prediction, mutation identification and comparative genome analyses. Overall, HBV genotypes B (44.1%) and C (46.2%) together with predicted serotypes *adr* (36%), *adw2* (29%) and *ayw1* (19.9%) were the most commonly circulating HBV types in the studied region. The three HBV variants identified most frequently were p.V5L, c.1896G>A and double mutation c.1762A>T/c.1764G>A, while genotypes B and C had the widest range of mutation types. The study also highlighted the distinct nucleotide diversity of HBV genotypes for whole genome and along the genome length. Therefore, this study provided a robust update to HBV currently circulating in Southeast Asia, Australia and New Zealand as well as an insight into the association of HBV genetic hypervariability and prevalence of well reported mutations.

## 1. Introduction

Hepatitis B virus (HBV) infection is a global health burden. 257 million people had chronic hepatitis B (CHB) infection worldwide in 2015 [1]. CHB can develop into liver cirrhosis, and hepatocellular carcinoma (HCC), resulting in 720,000, and 470,000, deaths worldwide in 2015, respectively [1].

HBV is a small, enveloped DNA virus of the *Hepadnaviridae* family. Its genome (3.2 kbp) is circular and partially double-stranded with four overlapping open reading frames (ORFs) encoding for reverse transcriptase (RT), pre-surface/surface (pre-S/S) proteins, precore/core (PC/C) proteins and transcriptional transactivator X protein [2]. The overlapping structure of pre-S/S and RT ORFs is essential for HBV surface antigen (HBsAg) antigenicity and RT functionality [3]. The PC/C region is responsible for formation of two polypeptides: (1) HBV core antigen (HBcAg)—a structural protein of the nucleocapsid translated from pregenomic RNA and (2) HBV e antigen (HBeAg)—a viral secreted protein translated from PC RNA [4,5]. The core promoter is located immediately upstream of the PC/C region and consists of the basal core promoter (BCP) and the upper regulatory region (URR). URR is composed of the negative regulatory element (NRE) and the core upstream regulatory sequence (CURS), with a possible role in regulating core promoter activity [6]. The X ORF encodes for X protein, which is associated with viral regulation, cell growth and carcinogenesis by disrupting several cellular signaling pathways [7,8,9].

Several factors contribute to HBV genetic diversity. HBV possesses a unique replication cycle via an RNA intermediate, using an error-prone RT enzyme [2,10,11]. This results in a nucleotide substitution rate of >10^−5^ substitution/site/year, which is higher than most DNA viruses and similar to RNA viruses with low mutation rate [2,10,11]. Moreover, it has a rapid replication rate with over 10^11^ virus particles released in circulation per day [12].

Due to its high sequence heterogeneity, HBV has 10 recognised genotypes (A to I, and a provisional genotype J) with at least 8% nucleotide divergence, and over 35 subgenotypes with at least 4–8% nucleotide divergence over the whole genome [13,14]. Viral strains typically exhibit geographical and ethnic distributions [13,15]. Generally, genotypes A and D are highly prevalent in Europe, North Africa, North America and Asia, whilst in East and Southeast Asia, Australia and New Zealand genotypes B and C are more commonly found [15]. Genotypes E and A predominate in Sub-Saharan Africa, whilst genotypes F, G and H are recorded in Latin America [15]. To date, only a single instance of genotype J was identified in Japan [16].

Recombination is common in HBV, classified into 44 different patterns where at least two different HBV genotypes or subgenotypes exchange genetic information [17,18]. For instance, subgenotypes B2, B3 and B4 are recombinants of genotypes B and C while fragments of genotypes A, C and G are observed in genotype I [2,19]. There is also evidence of recombinants C/D and D/E [20].

HBV can be categorised into different serotypes based on serological reactivity of HBsAg including *ayw1, ayw2, ayw3, ayw4, ayr, adw2, adw4, and adr*. The presence of subdeterminants, *d/y* and *r/w*, depends on variation at amino acid positions 122, 160, 127, 159 and 140 [21,22]. Although the same HBV genotypes can encode different serotypes and one serotype can belong to different genotypes, the distinct variants associated with viral serotypes are consistently found at the genomic level [21,23].

The genetic variability of HBV plays an important role in clinical outcomes and planning HBV prevention strategies. Acute genotype A infections persist longer than genotype D. Genotypes A and D surpass B and C in viral chronicity, possibly due to their different roles in host-viral interactions [24,25,26]. Genotype C infections have an increased risk of higher viral loads and development of cirrhosis and HCC than genotype B [24,27]. Interferon-based treatment was shown to be more effective for genotypes A and B than genotypes C and D, demonstrated through lower expression of HBsAg, HBeAg and decreased HBV DNA levels [28].

Most HBV vaccines are generated from genotype A2 or C strains [29,30]. The vaccines are protective against non-A/C genotype infections by largely boosting the production of antibodies targeted against determinant *a* at amino acid positions 121–149 which corresponds to the first loop of B-cell epitope of HBsAg for all HBV genotypes and serotypes [3,21,30,31]. However, cases of cross-genotype HBV infection of vaccinated individuals have been recorded, raising questions as to the complete protection of HBV immunoprohylaxis, especially in countries where genotypes A and C are not prevalent [32,33,34].

Several HBV mutations within particular coding regions are associated with important clinical manifestations. Mutations across the major hydrophilic region (MHR) of HBsAg, where determinant *a* is located, disrupt the conformation of surface antigen, affect protein hydrophilicity and alter the binding of neutralizing antibodies. This allows HBV to escape host immune responses for clearance [35,36], resulting in failed immunoprophylaxis [37,38,39,40,41,42]. RT mutations are involved in the development of drug-resistant viral strains and loss of efficacy of nucleos(t)ide analogs and anti-viral drugs. These include p.M204IVS, p.V173L, p.A181T and p.L80VI for lamivudine, p.N236T and p.181V for adenovir, p.A194T for tenofovir, and p.M250V/p.I169T and p.T184G/p.S202I for entecavir [43,44]. Mutations within the PC region acquired after HBeAg seroconversion (c.1896G>A and p.P130T) can lead to defective HBV replication, decreased or lost secretion of HBeAg, HBeAg-negative CHB and even HCC with poorer prognosis [5,7,45,46,47]. The two HBcAg mutations p.I/F97L and p.P130T are frequently observed concurrently in CHB patients [44], leading to the secretion of virions containing immature genomes and hypermaturations, respectively [48]. Meanwhile, double mutations c.1762A>T / c.1764G>A in the BCP region and mutation c.1613G>A within NRE, are associated with the suppressed expression of HBeAg, increased viral load in HBV carriers and increased risk of liver carcinogenesis [6,49,50]. A study on HBV X mutations reported the presence of the mutant p.V5M/L amongst Korean patients with severe liver diseases such as cirrhosis and HCC [51].

The rate of international travel to Southeast Asia, Australia, New Zealand has increased in recent years. Given the geographically distinct distribution of HBV genotypes and serotypes, this trend could influence the prevalence of HBV strains currently circulating in this region. Moreover, there have been conflicting reports of the predominant genotype in a number of countries within this area. A study in 2004 [13] identified only HBV genotypes B and C in the region, except for Australia, in which genotypes D and C were found. Meanwhile, a recent systematic literature review [15] reported genotypes B, C, D and A circulating in these countries. This study therefore aimed to robustly update the situation of HBV types in this region. Moreover, given the potential complications arising from aforementioned HBV mutations, we sought to estimate a number of mutations across the viral genome by genotypes. We also investigated the nucleotide variability of HBV genotypes in the same region to explore whether there is an association between HBV heterogeneity and mutation rates for each genotype.

## 2. Materials and Methods

### 2.1. Data Collection from NCBI Nucleotide Database

A search for human HBV complete genomes listed on the NCBI nucleotide database was conducted in August 2019, firstly using the term [(“HBV” OR “hepatitis B virus”) AND “complete genome”]. We then applied the following filters: viruses for species, gDNA/RNA for molecule types, International Nucleotide Sequence Database Collaboration (INSDC) (GenBank) and [country name]. The studied countries were located in Southeast Asia, Australia and New Zealand, with names as described by the United Nations (https://unstats.un.org/unsd/methodology/m49/). Each genome was manually checked for inclusion, including HBV complete genome, *Homo sapiens* host, country area, and unique sequences. The country of origin of each genome was checked either by using the information given on NCBI or listed in publications.

### 2.2. Analysis of HBV Genotypes

For each sequence, we used NCBI HBV genotyping [52] to assess the agreement between genotypes submitted on GenBank and results provided on the HBVdb genotyping tool [53]. We included sequences KF214650|I, EU833891|I, JF899337|I, and AB486012|J for genotypes I and J as references. A genotype was assigned based on (1) concurrence of genotype assigned by three sources (Genbank submission, NCBI HBV genotyping and HBVdb genotyping tool) or (2) the first E-value given by NCBI BLAST < 1 × 10^−75^ in the case of genotype assignment disagreement from one or more sources/tools. We nominated unverified genotypes or potential recombinants by phylogenetic analysis using FastTree 2.1.11 [54,55] and RAxML 8.2.11 [56]. For FastTree, the default tree building setting was used. For RaxML, we used a general time-reversible (GTR) GAMMA I model, rapid bootstrapping and 1000 replicates. Figure 1 illustrates the steps of genotyping the collected sequences. Sequences were aligned using MAFFT [57,58] and Geneious Prime 2019.2.1 for the whole study.

### 2.3. HBV Recombination Analysis

We utilised the Recombination Detection Program (RDP) 4 [59] to detect potential recombination events within our data. Five built-in methods (RDP [60], Geneconv [61], Maxchi [62], Chimaera [63] and 3SEQ [64]) were run iteratively with their default settings, to identify potential recombinants by each method. Sequences detected as potential recombinants with overlapping breakpoints by all 5 methods were then pooled together with the sequences of unverified genotypes/recombinants for the phylogenetic analysis as outlined in 2.2. Reference sequences were included (Appendix A).

### 2.4. Representative Phylogenetic Analysis

A number of sequences were randomly selected from the complete dataset for the identified genotypes to establish a representative phylogeny. The number of sequences chosen for the representative phylogeny was proportional to the number of sequences of each genotype in our dataset: 5 sequences for genotype A, 20 for B, 20 for C, 10 for D, 1 for G and 5 for I. Potential recombinants were also included in the population. Approximated maximum-likelihood trees were built by using FastTree 2.1.11 [54,55], RAxML 8.2.11 [56] and PHYML 3.3.20180621 [65]. Model parameters for FastTree and RAxML were as described in 2.2, Model parameters for PHYML were established by using jmodeltest: GTR+I+G, proportion of invariable sites 0.463, gamma distribution of 0.854 and 1000 bootstraps. Reference sequences were included (Appendix A).

### 2.5. HBV Serotype Prediction

Serotypes for each HBV genome were predicted using the decision tree developed by Purdy et al. (2007) [22]. Translated HBsAg sequences were analysed with the mutation reporter tool [66] to determine serotypes. We used FLLVLLDY as an input for anchor motif at amino acid position 93. We confirmed our predictions by using the HBV serotyper tool (http://hvdr.bioinf.wits.ac.za/SmallGenomeTools/) [67] or manually. Fourteen sequences that were not predicted to any serotypes by either of the two methods above were individually investigated for their amino acid composition.

### 2.6. Nucleotide Diversity

We used DnaSP 6 for a comprehensive analysis of sequence variation by each genotype [68]. All degenerate nucleotides were converted to Ns prior to analysis as required by the software. Sites with gaps in the complete data file were excluded from the analysis. To evaluate the degree of polymorphism within genotypes, we estimated the average number of nucleotide differences per site between two sequences, called nucleotide diversity Pi (μ), with window length of 100 bp and step size of 25 bp.

### 2.7. Analysis of Nucleotide and Amino Acid Variations

Analysis of HBV variations was performed across five regions—NRE, BCP, PC, C, HBsAg, polymerase/RT, and X. Variations selected for analysis were all reported to be associated with viral life cycle and/or clinical complications (Figure 2) [69,70]. Drug resistance mutations over the RT ORF and escape mutations within HBsAg ORF were identified by HIV-grade HBV drug resistance interpretation (DRI) [70]. Other variants were determined based on nucleotides or amino acids substituted at the selected positions along nucleotide sequences or translated sequences using Geneious Prime 2019.2.1.

## 3. Results

### 3.1. Data Collection from NCBI Nucleotide Database

Of the 9182 complete HBV genomes identified on NCBI nucleotide database, 930 genomes met the inclusion criteria (File S1 and Figure 3). Five pairs of HBV genomes, despite having different GenBank accession numbers, were identical, therefore, one sequence of each pair was excluded. No human HBV complete genomes were found for the following geographical locations: Brunei Darussalam, Singapore, Timor-Leste, Christmas Island, Cocos Islands, Norfolk Island, Heard Island and McDonald Island. No relevant information regarding clinical condition was available for any of the sequences collected.

### 3.2. Analysis of HBV Genotypes

Over 50% of sequences originated from Malaysia and Vietnam, with 296 and 180 sequences, respectively. We identified six HBV genotypes (A, B, C, D, G and I) circulating in Southeast Asia, Australia and New Zealand (Table 1 and Figure 4). The frequencies of each HBV genotype varied widely between countries. In the studied region, genotypes C and B were the most prevalent, constituting 46.2% and 44.1%. Genotype I (2.3%) was observed only in Laos, Vietnam and Thailand, whilst genotype D (5.3%) was found in New Zealand, Indonesia and Malaysia. Only one sequence was genotyped G and originated in Thailand. Countries with the widest range of circulating HBV genotypes were Malaysia (A, B, C and D) and Thailand (B, C, G and I), followed by Vietnam (B, C and I), Philippines (A, B and C) and Indonesia (B, C and D).

### 3.3. HBV Recombination Analysis

We ran five automatic detection methods in RDP4 to detect potential recombination within our data (Table 2). In total, 65 genomes were identified as recombinant by all five methods; 35 of them had overlapping breakpoint positions, including 10 with the same major parents who contribute greater proportion of sequence. The 65 suspected recombinants were pooled together with sequences of ambiguous genotype or suspected recombination for further analysis with FastTree and RaxML (Appendix A). According to these phylogenies, 34 of the 65 sequences were clustered in genotype B, particularly B2 and B3, whilst 13 clustered in genotype C and 12 in genotype A. Of the seven sequences that were initially suspected as B/BC, C/CB or G/CG recombinants (Table 1) by HBV genotyping, two C/CB were located between the clusters of genotypes B2 and C, two G/GC between G and C, while the three B/BC were positioned within the cluster of B2 and B3. Results of recombination and phylogeny analyses are summarised in Appendix A. They all shared approximately 92% similarity with their corresponding reference sequences.

### 3.4. Representative Phylogenetic Analysis

Generally, all three trees built with FastTree, RAxML and PHYML (Appendix A) were consistent in clustering the sequences of the same genotypes identified in 3.2, supporting our genotyping results. The divisions into genotypes A, B, D and I were highly supported by all three methods. Lower support values were calculated for genotype G (82% by RAxML and 78.1% by PHYML) and C (62% by RAxML and 53.5% by PHYML). PHYML and RAxML trees were generally congruent, except for five sequences, one of genotype B and four of genotype C, though still clustering within clades of their corresponding genotypes. Differently, for FastTree tree, genotype roots were relocated and a bigger number of sequences of genotypes B, C, D and F changed their arrangement within the same clade.

### 3.5. HBV Serotype Prediction

We predicted serotypes for the 930 HBV genomes using the decision tree of Purdy et al. [22]. Table 3 summarises the distribution of the predicted HBV serotypes by genotypes. Serotypes *adr, adw2* and *ayw1* were most prevalent in the studied region, accounting for 36%, 29% and 19.9%, respectively. They were predominantly found with HBV genotypes B and C, which are common in Southeast Asia and Australia. All sequences of serotype *adr* and *ayr* were within genotype C. Serotypes *adw2* and *ayw1* were found in a wider range of genotypes than the other serotypes. Serotypes *adw3, ayw2* and *ayw3* were scattered among genotypes B, C and D with no discernible distribution pattern.

### 3.6. Nucleotide Diversity

Nucleotide diversity analysis indicates a wide range of genetic variability for whole genome (Table 4) and along the HBV genome length (Figure 5). Genotype G was not included because only one sequence of this genotype was identified. Genotypes B and C demonstrated a higher number of segregating sites, Pi values and average number of nucleotide differences, indicating greater genetic diversity compared to other genotypes (Table 4). Despite the lowest number of genomes included in the dataset, genotype A demonstrated Pi and k values greater than genotypes D and I. This suggests that genotype D and I were the most conserved in their overall nucleotide sequences.

Moreover, we found genetic differences appeared to be unevenly distributed along the genome length (Figure 5). At a genotypic level, genotypes B and C were the most diverse within the overlapping RT and pre-S/S coding genes, while genotype A expressed more genetic variations located near or within X gene and PC/C gene. Genotypes D and I were more conserved than the others over the whole genome, except for a distinct locus in RT ORF.

### 3.7. Analysis of Single Nucleotide and Amino Acid Variations

Our dataset was investigated for the frequency of the selected variations by genotype (Table 5). We observed the p.V5L variation was present in almost all sequences analysed (≥83% prevalence for all genotypes). c.1896G>A was the second most common variant, particularly with HBV genotypes B, D, I and C, while c.1762A>T/c.1764G>A ranked third but was less genotype-specific. It was noted that the compound nucleotide substitution at positions 1762 and 1764 within BCP was more likely to occur together, rather than to have either substitution singularly. Escape mutations in the viral surface coding gene (14.4%) and c.1613G>A in NRE (12.8%) also occurred frequently, predominantly in genotypes B and C. Additionally, p.P130T in core gene and p.V5M in the X gene were mainly acquired in genotypes B and C, while drug resistance mutations occurred sporadically for genotypes A–D.

## 4. Discussion

This *in silico* study of the distribution of HBV genotypes and recombinants in Southeast Asia, Australia and New Zealand was performed through the phylogenetic, recombination and comparative genomic analysis of 930 human HBV complete genomes. We observed a high prevalence of genotypes C and B in the studied region. This observation is consistent with previous studies [13,15]. The exception is Laos, where we identified genotype I as the dominant genotype, rather than genotypes B and C as reported by Norder [13] and Velkov [15]. We also detected frequent recombinants between genotypes B and C, all from Thailand and Malaysia.

This study also investigated the distribution of HBsAg subtypes in the study region predicted using a decision tree [22]. The three most prevalent serotypes were *adr* (36%), *adw2* (29%) and *ayw1* (19.9%). We found serotypes *adr*, *ayr* and *ayw3* most commonly associated with genotype C, *adw2* for B and C, *adw3* and *ayw1* for B, and *ayw2* for D. These findings differ to Norder’s study [13], which reported that the serotypes *adw2*, *ayw1* and *ayw3* were more pronounced amongst genotypes A, B and D. Moreover, our observations of HBsAg subtype distribution for genotypes B and D are dissimilar to the finding of Chen et al. [71] who reported that *adr* and *ayw2* were prevalent among genotype B strains while most of genotype D sequences were assigned to *adr* [71]. This is likely due to the geographical selection of HBV sequences used in this study. We targeted Southeast Asia, Australia and New Zealand, while the Norder study [13] was worldwide and the Chen study [71] focused on China.

To our knowledge, this study is also the first to investigate nucleotide diversity between HBV genotypes. We found that levels of nucleotide diversity varied amongst the different genotypes. Genotypes C and B showed an overall higher genetic diversity and more mutation types than genotypes A, D and I. Furthermore, we found that each genotype demonstrated varying genetic diversity within different ORFs (Figure 5). Specifically, genotypes B and C displayed the highest heterogeneity in the polymerase and pre-S/S regions while genotype A showed the highest diversity within the X and PC/C genes.

Information on the distribution of HBV mutations along genome length amongst genotypes is limited. Our study showed that the frequency of these mutants varied with genotypes (Table 5). This observation was also reported by Bayliss et al. [69]. We found that sequences with mutations, except for c.1896G>A, were mainly associated with to genotype C. However, we observed that p.V5L, c.1896G>A and c.1762A>T/c.1764G>A were the three most prevalent variants, contrasting to c.G1613A, c.1762A>T/c.1764G>A and c.1896G>A in the Bayliss study. These differences are likely attributable to sample selection criteria. In the Bayliss study, only HBeAg-positive patients with CHB were included, which may favour the presence of particular mutations. In our study, there was very little information linked to clinical diseases available on the NCBI nucleotide database for the collected genomes. Interestingly, in our study, escape mutations associated with risk of failed immunoprophylaxis [37,38,39,40,41,42] occurred significantly for genotypes A and C (totally 18%), compared to non-A/C genotypes (totally 10.6%) (X^2^ = 10.588, df = 1, *p* = 0.001138), given that most currently used HBV vaccines are generated from genotype A or C strains [29,30]. Due to the unavailability of relevant clinical information for these entries in GenBank, vaccine status for these sequences is unknown.

In considering whether an association between genetic hypervariability and mutation rate exists for each HBV genotype, we observed that the PC/C ORF of genotype A demonstrated high nucleotide diversity had a high proportion of c.1762A>T/c.1764G>A (42%) and c.1613G>A (16%) mutations. By contrast, although the polymerase and pre-S/S genes of genotypes B and C demonstrated the highest diversity compared with other ORFs, the percentage of mutations within these regions, including drug resistance and escape mutations, were much lower than all other mutations within the PC/C region combined (Table 5). This observation may be explained by a finding of Osiowy et al. [10], that the RT region has a higher ratio of synonymous to non-synonymous substitutions (dS/dN) than the core coding region. In other words, nucleotide subtitutions in the RT ORF are less likely to alter amino acid sequence, and therefore are less likely to be associated with clinically significant mutations than variation in the core region. This feature may allow HBV to play critical roles of the overlapping pre-S/S and RT regions on maintaining HBV transcription and HBsAg functionality against host immune system [3].

We recognise that our study has limitations. Our sequence selection was regionally focused, and HBV strains circulating in countries outside our studied area could account for genetic diversity of genotypes and subgenotypes which are absent or less frequent in Southeast Asia, Australia and New Zealand. There is a need for a global study of HBV genetic variability, to better understand contemporary distribution pattern—particularly considering the rise in human migration and ease of global travel.

In conclusion, this investigation is an up-to-date, thorough report on the distribution of HBV types circulating in Southeast Asia, Australia and New Zealand. We highlighted HBV nucleotide variability, and the prevalence of different mutations within genotypes and ORFs. Given that several HBV variants are associated with failure of immunoprophylaxis, drug resistance and liver disease progression, the understanding of variability and mutations within genotypes highlighted in our study may guide clinicians in predicting the risks associated with failed protection and unsuccessful treatment.

## Figures and Tables

**Figure 1 viruses-12-00427-f001:**
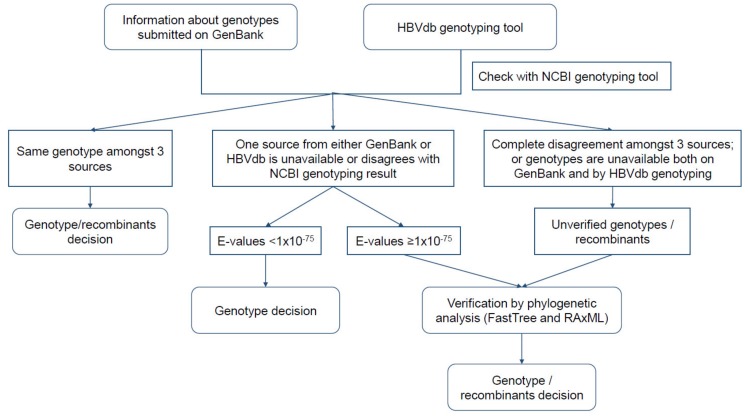
Flowchart of genotype assignment for each sequence retrieved from the NCBI nucleotide database.

**Figure 2 viruses-12-00427-f002:**
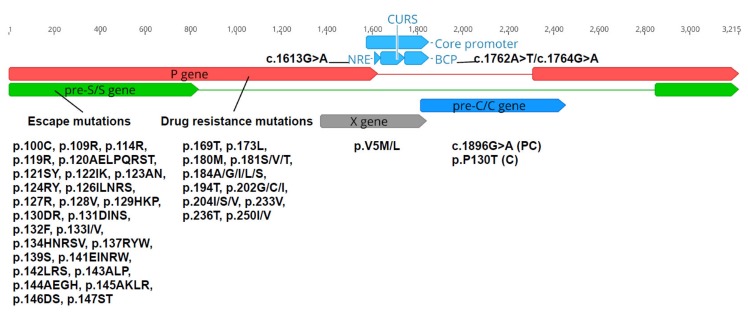
Selected mutations across RT, HBsAg, NRE, BCP, PC, C and X ORFs for analysis. Nucleotide position 1 on an HBV genome corresponds to the HBV EcoR1 restriction site.

**Figure 3 viruses-12-00427-f003:**
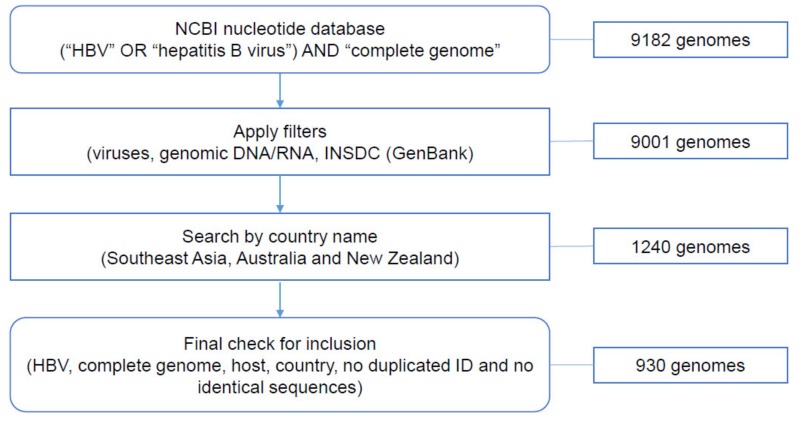
Number of HBV complete genomes retrieved through each step of data collection.

**Figure 4 viruses-12-00427-f004:**
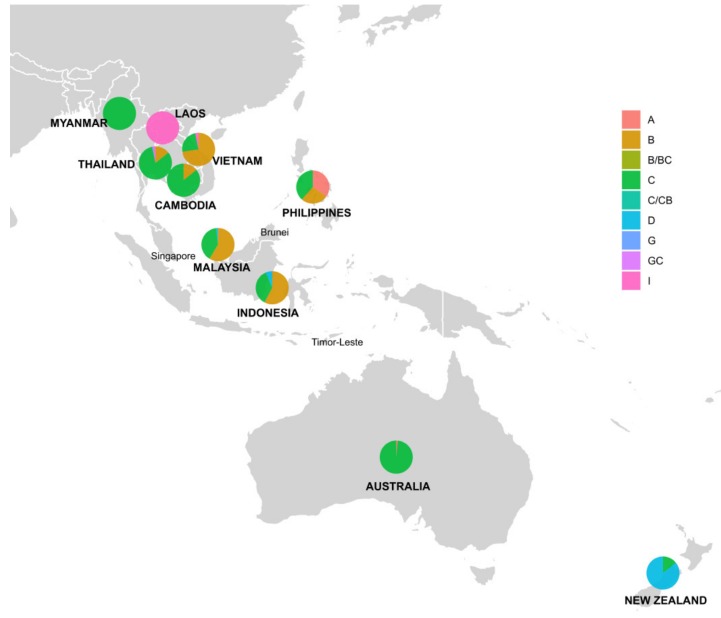
Distribution of HBV genotypes and recombinants by country in Southeast Asia, Australia and New Zealand. Pie charts indicate the percentage of HBV genotypes with the diameters. Countries with human HBV complete genomes collectable from NCBI nucleotide database were highlighted in bold and uppercase.

**Figure 5 viruses-12-00427-f005:**
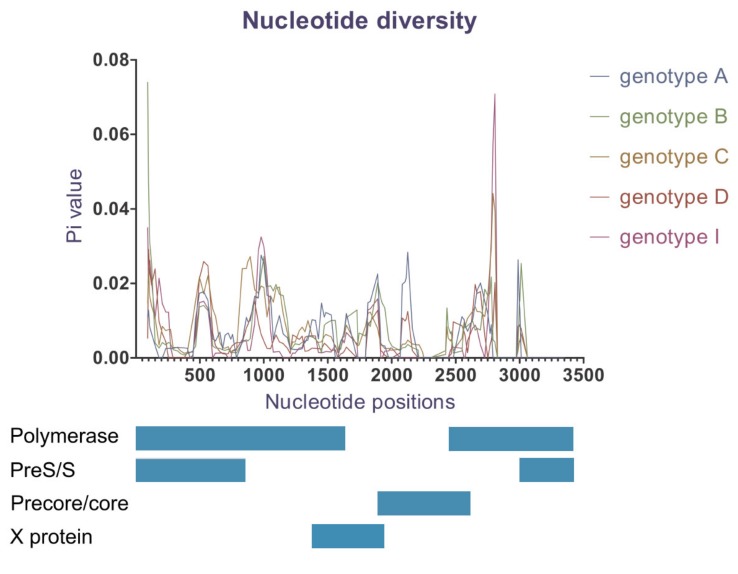
Nucleotide diversity (Pi) over HBV genome. The MAFF-aligned genomes had 3434 bp in length; genotypes were colour-coded (sites with gaps in the complete data file were excluded; window length 100 bp, step size 25 bp).

**Table 1 viruses-12-00427-t001:** Identified HBV genotypes by country.

	Genotype	A	B	C	D	G	I	B/BC or C/CB	G/GC	Total
**Country**										
Australia		1		66						67 (7.2%)
Cambodia			4	24						28 (3%)
Indonesia			82	51	9					142 (15.3%)
Laos							14			14 (1.5%)
Malaysia		2	170	116	4			4		296 (31.8%)
Myanmar				18						18 (1.9%)
New Zealand				6	36					42 (4.5%)
Philippines		9	7	10						26 (2.8%)
Thailand			16	96		1	1	1	2	117 (12.6%)
Vientam			131	43			6			180 (19.4%)
**Total**		12 (1.3%)	410 (44.1%)	430 (46.2%)	49 (5.3%)	1 (0.1%)	21 (2.3%)	5 (0.5%)	2 (0.2%)	930 (100%)

**Table 2 viruses-12-00427-t002:** HBV recombination detection by five automatic methods built in RDP4.

	Detection Method	RDP	3SEQ	Chimaera	Geneconv	Maxchi
Events/Recombinants						
Recombination events		83	80	49	87	41
Potential recombinants (possibly present in different events)		467	613	744	1199	1355
Potential recombinants (after removing replicate recombinants)		380	566	510	857	548
Recombinants identified by all methods: 83 (65 sequences from our data and 18 reference sequences)						
+ with overlapping breakpoint positions: 35 (31 sequences from our data and 4 reference sequences)						
+ with at least one same major parent: 10 (8 sequences from our data and 2 reference sequences)						

**Table 3 viruses-12-00427-t003:** Predicted HBV serotypes by genotype and country in Southeast Asia, Australia and New Zealand.

	Serotype	*adr* (*n* = 335, 36%)	*adw2* (*n* = 270, 29%)	*adw3* (*n* = 7, 0.8%)	*ayr* (*n* = 6, 0.7%)	*ayw1* (*n* = 185, 19.9%)	*ayw2* (*n* = 45, 4.8%)	*ayw3* (*n* = 70, 7.5%)	Unassigned (*n* = 12, 1.3%)
Genotype/ Country									
***A***			***9***			***2***			***1***
Australia			1						
Malaysia			1						1
Philippines			7			2			
***B***			***218***	***5***		***172***	***1***	***6***	***8***
Cambodia			2			2			
Indonesia			46	1		34			1
Malaysia			141	4		20	1		4
Philippines			1			6			
Thailand			12			4			
Vietnam			16			106		6	3
***C***		***331***	***29***	***1***	***6***			***60***	***3***
Australia		6						60	
Cambodia		23	1						
Indonesia		45	3		3				
Malaysia		107	5	1					3
Myanmar		17	1						
New Zealand		6							
Philippines		1	9						
Thailand		86	7		3				
Vietnam		40	3						
***D***						***1***	***44***	***4***	
Indonesia						1	8		
Malaysia								4	
New Zealand							36		
***G***			***1***						
Thailand			1						
***I***			***11***			***10***			
Laos			4			10			
Thailand			1						
Vietnam			6						
***B/BC***		***1***	***1***	***1***					
Thailand				1					
Malaysia		1	1						
***C/CB***		***1***	***1***						
Malaysia		1	1						
***G/GC***		***2***							
Thailand		2							

**Table 4 viruses-12-00427-t004:** DNA polymorphism of HBV genotypes identified in Southeast Asia, Australia and New Zealand.

Genotype	Number of Polymorphic (Segregating) Sites	Average Number of Nucleotide Differences (k)	Nucleotide Diversity (Pi)
Genotype A (*n* = 12)	44	10.788	0.00775
Genotype B (*n* = 410)	454	11.515	0.00827
Genotype C (*n* = 430)	446	12.548	0.00901
Genotype D (*n* = 49)	68	7.770	0.00558
Genotype I (*n* = 21)	37	8.148	0.00585
Total *N* = 930 (including genotype G and recombinants)	676	20.944	0.01505

**Table 5 viruses-12-00427-t005:** Number and percentage of selected mutations by HBV genotype ^1^.

Mutation Type (Region)	A (*n* = 12)	B (*n* = 410)	C (*n* = 430)	D (*n* = 49)	G (*n* = 1)	I (*n* = 21)	B/BC, C/CB, G/GC (*n* = 7)	Total (*n* = 930)
Drug resistance mutations (polymerase)	1 (8%)	15 (3.7%)	22 (5%)	2 (4%)	0	0	0	40 (4.3%)
Escape mutations (pre-S/S)	2 (16%)	41 (10%)	79 (18%)	8 (16%)	0	2 (10%)	2 (27%)	134 (14.4%)
c.1613G>A (NRE)	2 (16%)	44 (11%)	67 (15%)	0	1 (100%)	4 (19%)	1 (14%)	119 (12.8%)
Only c.1762A>T (BCP)	0	7 (2%)	3 (0.7%)	0	0	0	0	10 (1.1%)
Only c.1764G>A (BCP)	1 (8%)	3 (0.7%)	20 (5%)	1 (2%)	0	0	0	25 (2.7%)
Double c.1762A>T/ c.1764G>A (BCP)	5 (42%)	60 (15%)	159 (37%)	7 (14%)	1 (100%)	5 (24%)	2 (27%)	239 (25.7%)
c.1896G>A (PC)	0	193 (47%)	77 (18%)	13 (27%)	0	4 (19%)	2 (27%)	289 (31.1%)
p.P130T (C)	0	28 (6.8%)	60 (14%)	0	1 (100%)	0	0	89 (9.6%)
p.V5M (X)	0	8 (2%)	11 (2.6%)	0	0	0	1 (14%)	20 (2.2%)
p.V5L (X)	10 (83%)	401 (97%)	395 (92%)	48 (97%)	1 (100%)	21 (100%)	6 (86%)	882 (95%)

^1^ Note: One genome can acquire one or more variations.

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
