# Peer review of "In Silico Analysis of Genetic Diversity of Human Hepatitis B Virus in Southeast Asia, Australia and New Zealand"

_viruses, 2020, doi:10.3390/v12040427_

Round 1
Reviewer 1 Report
The authors did a comprehensive analysis of HBV sequences in Southeast Asia, Australia, and New Zealand. They reported the associations between sequence diversity and predicted HBV serotypes. In general, the rationale is straightforward and the authors nicely described their appearances, which makes readers are easily to repeat their procedures. The results look interesting and revealed some mutation hotspots, which deserve further investigations. I only have few suggestions. 1. What is the reason to select the interested regions/countries in this paper? It is not clearly described. As noted by the authors in discussion, HBV is spreading across whole world and thus it is somewhat weird to only focus on some specific regions without a well-described reason. 2. I am impressed by the authors to annotate drug-resistant variants. However, the descriptions are not complete. Are the drugs all the same across all loci? Do we have any concern about drug dosages? The authors shall provide more details. 3. Do the authors suggestion any linkage between predicted HBV seroytypes and drug efficacy? Can the authors discuss this?Author Response
Response to Reviewer 1 Comments
Point 1: What is the reason to select the interested regions/countries in this paper? It is not clearly described. As noted by the authors in discussion, HBV is spreading across whole world and thus it is somewhat weird to only focus on some specific regions without a well-described reason. 

Response 1:
The reason we selected Southeast Asia, Australia and New Zealand for this paper is because the rates of international travel to these countries have been increasing in recent years, and so we were interested in investigating if the current HBV strains in circulation are different to what has been reported in the past. Given the geographically distinct distribution of HBV genotypes and serotypes, increased travel from other parts of the world could influence the prevalence of HBV strains currently circulating in this region. Moreover, there was conflicting evidence of genotypes predominating in a couple of countries in the area. A study in 2004 (Intervirology 2004, 47, 289–309) identified only HBV genotypes B and C in the region, except for Australia with identified genotypes D and C. Meanwhile, a recent systematic literature review (Genes 2018, 9, 495; doi:10.3390/genes9100495) reported genotypes B, C, D and A circulating in these countries. With this in mind, we thought it was necessary to focus on HBV strains currently circulating in Southeast Asia, Australia and New Zealand.
Modifications have been made to the introduction (line 106 – line 114).
Point 2: I am impressed by the authors to annotate drug-resistant variants. However, the descriptions are not complete. Are the drugs all the same across all loci? Do we have any concern about drug dosages? The authors shall provide more details.
Response 2:
Due to the unavailability of relevant clinical information for the entries in GenBank, we were unable collect data on drug use, dosage, or clinical consequences for the studied sequences. This is why we did not analyse the association between the selected variants and drug dosage or efficacy in our in silico analysis of HBV genomes. A clinical study on HBV drug-resistant variants by (sub)genotypes would be interesting but is beyond the scope of this study and the data analysed here.
Point 3: Do the authors suggestion any linkage between predicted HBV seroytypes and drug efficacy? Can the authors discuss this?
Response 3:
Similar to our response to point 2, due to the unavailability of relevant clinical information for the entries in GenBank, we were unable to collect relevant clinical information in terms of drug use, dosage and treatment efficacy. Therefore, we were not able to perform an analysis on the association between predicted HBV serotypes and drug efficacy. Clinical data is required for such an analysis.
Reviewer 2 Report
The paper entitled "
In silico analysis of genetic diversity of human hepatitis B virus in Southeast Asia, Australia and New Zealand" is exploring HBV variability related to genotypes in a localized area in the world in a very systematic manner and with deep analysis. It is bringing new insight for people in the field and coulb be extent to other genotypes and areas. One could regret no mention of the variability in term of sub genotypes and possibly to discuss the clinical consequences that are associated with more or less variability depending on the genotypes and genes.
Author Response
Response to Reviewer 2 Comments
Point 1: … no mention of the variability in term of sub genotypes and possibly to discuss the clinical consequences that are associated with more or less variability depending on the genotypes and genes.
Response 1:
We agree with the reviewer that it may be beneficial to study the HBV variability related to subgenotypes. However, no HBV subgenotyping tools are currently available. Information about HBV subgenotypes submitted to GenBank is widely missing for the collected HBV complete genomes, compared to HBV genotypes. Moreover, we designed our study to maximise the value of using HBV genotype information submitted to Genbank and the two HBV genotyping tools in combination with phylogenetic analysis of a representative population. This allowed us to confidently determine genotypes for the collected sequences.
We also acknowledge that discussion on the association between HBV genetic diversity and clinical consequences is interesting. Unfortunately, due to the unavailability of relevant clinical information for the entries in GenBank, we were unable to do such analyses.